# Physical Activity in the COVID-19 Era and Its Impact on Adolescents’ Well-Being

**DOI:** 10.3390/ijerph20043275

**Published:** 2023-02-13

**Authors:** Elena Bozzola, Sarah Barni, Andrea Ficari, Alberto Villani

**Affiliations:** Pediatric Unit, IRCCS Bambino Gesù Children’s Hospital, 00100 Rome, Italy

**Keywords:** adolescent, physical activity, exercise, COVID-19

## Abstract

Physical inactivity and sedentary habits are among the major risk factors for decreased physical and mental well-being. Since the onset of the COVID-19 pandemic, normal daily routines changed, including physical activity (PA) habits. The aim of this manuscript is to review the literature according to the PRISMA guidelines in order to analyze the changes in PA and exercise practice after the onset of the COVID-19 pandemic and its impact on the well-being of adolescents. A PubMed search was performed using the keywords “Exercise” [Mesh]) AND “COVID-19” [Mesh], and filters to limit the research to pertain to adolescents (13–18 years) and English reports. Out of the search, 15 reports met the criteria for inclusion in the study. The main findings outlined a global decrease in PA levels associated with decreased well-being levels, modified eating habits and leisure time activity, and increased obesity, anxiety, and depression among adolescents. PA is a significant health determinant and should be improved through the awareness of the benefits of regular PA and of the risks of sedentary behavior, as well as through support from family, friends, and teachers. Providing PA at school, as a part of the academic program, increasing the availability of equipment and facilities, and promoting at-home PA options are suggested as support for increasing PA in all countries and settings.

## 1. Introduction

The World Health Organization defines physical activity (PA) as any movement of the body produced by skeletal muscles that requires energy expenditure [1,2]. This term encompasses both structured activities, such as sports, and leisure time movements, such as walking or running. According to the WHO definition, physical inactivity is defined as an insufficient level of physical activity to meet the present physical activity recommendations. For children and adolescents, sedentary behavior includes time spent sitting or lying down with low energy expenditure, while awake, in the context of educational, home, and community settings and transportation [2]. PA may support both physical and mental well-being in adolescents [3,4]. In adolescence, PA may improve physical fitness, cardiometabolic and bone health, cognitive outcomes (academic performance and executive function), and mental health as well as reduce adiposity through biological, psychosocial, and behavioral mechanisms [5]. In contrast, sedentary habits are related to poor health outcomes, including higher weight gain and cardiometabolic risk, lower psychological health levels, and sleep disturbances [3]. International guidelines promote moderate-to-vigorous physical activity (MPVA) in children and adolescents, but a low percentage of youth meet the WHO recommendations of being active for at least 1 h every day of the week [5,6,7]. However, a major public health concern is that 81% of adolescents do not have adequate PA levels globally [8]. In particular, adolescence is a vulnerable period for regular PA as evidence suggests that the peak of PA levels occurs at the age of 13 and then declines by 7% per year thereafter [9]. Of note, during the COVID-19 pandemic, both organized activities and unstructured ones decreased, leading to increased inactivity and sedentary habits. Evidence confirms that during the pandemic, adolescent depression and anxiety increased while social support and well-being decreased [10,11]. Even if the causality of the relationship between well-being and PA may just be assumed, growing evidence postulates a positive effect of PA on mental health.

The purpose of this study was to analyze the changes in PA and exercise practices after the onset of the COVID-19 pandemic and their impact on the well-being of adolescents.

## 2. Materials and Methods

For the purpose of the study, a literature review was performed according to the PRISMA guidelines [12]. A search of the literature was undertaken through the electronic database PubMed, on 2 December 2022. For the aim of the study, the keywords used were (“Exercise” [Mesh]) AND “COVID-19” [Mesh], and filters were added to limit the research to pertain to adolescents (13–18 years) and English reports. Then, the research was downloaded and analyzed.

### 2.1. Eligibility Criteria

Reports on the physical activity (PA) of adolescents (13–18 years old) during the COVID-19 pandemic lockdown were eligible for inclusion in this review. The following were the inclusion criteria: − Adolescents aged between 13 and 18;− Full-length articles;− Language used: English;− Available data reporting PA before and during the COVID-19 pandemic.

The Exclusion criteria were:− Reports not pertinent to the field of investigation;− Reports without sample size’s age range;− Reports involving a selected group (i.e., with chronic disease);− Reports without data.

### 2.2. Selection Process

The study selection process was conducted based on the PRISMA flow diagram [13].

To reduce errors and bias, two authors independently analyzed the titles and abstracts produced by the research in order to find those eligible for inclusion and to discard those clearly irrelevant for the purpose of the review.

Afterward, full texts were retrieved and assessed for eligibility by the two screening authors. Then, following PRISMA guidelines, important references not included in the original search but relevant to the review were examined.

Finally, a third author double-checked the suitability of the selected reports.

Disagreements regarding inclusion/exclusion were settled through discussion between the researchers.

### 2.3. Data Collection Process and Data Items

Data was gathered in a Microsoft Excel spreadsheet to evaluate the differences in PA during the COVID-19 pandemic lockdowns compared to before that time.

The information extracted from the full-text reports included methodological, demographic, and outcome data: general information (authors, setting, and year), demographics (sample size and age group), research methods (aim and study design), and outcomes (statistics and findings).

### 2.4. Data Synthesis

Using the information gathered from the included studies, a scoping review was developed. The characteristics of the included studies were reported using descriptive statistics. No meta-analysis could be made with statistical work because of the variability of the studies in the assessment of PA and the quality of the available literature, i.e., data were too heterogeneous to be compared.

### 2.5. Synthesis Method

A descriptive synthesis of the general features of the included studies was provided. These characteristics include the author, setting and year of publication, type of study, participant characteristics, and findings. These results were then elaborated on in the discussion.

## 3. Results

The search of the selected electronic database produced n. 314 studies, including articles and reviews. One foreign language work was identified and discharged. Then, according to PRISMA guidelines, all abstracts were analyzed, and 202 records were excluded because they dealt with other topics not pertinent to this review, were based on an adult population, included only children and not adolescents, were not compared with the COVID-19 pandemic, or were case reports. Therefore, 111 records were eligible to be analyzed by reading their full-length text; however, 6 articles could not be retrieved. Therefore, 105 full-length reports were assessed for eligibility, and 92 were excluded because the majority did not describe age subgroups or they did not display any data, few included ≤12 or >18 years old within the adolescents or there were no age limits stated, and on one occasion, the study reported had already been described in another article. Finally, two relevant reports cited in other studies were added to this research.

In conclusion, 15 records were included in the revision [14,15,16,17,18,19,20,21,22,23,24,25,26,27,28].

Figure 1 presents the flow chart of the selection process, adapted from the PRISMA guidelines [13].

Table 1 shows the main findings and general features of the reports included in the literature review (Table 1).

The main findings of the literature review indicated a global decrease in PA, regardless of continent and country, which was associated with altered well-being levels, eating habits, and leisure time activity. (Figure 2) The main consequences were reported on weight and mood health, with evidence of increased obesity and internalizing symptoms (including anxiety and depression) among adolescents.

## 4. Discussion

Even before the onset of the pandemic, adolescents in many countries did not meet the WHO guidelines on PA [19,20,23]. After reviewing the literature, we discovered that from the onset of COVID-19, adolescents were even less active, most likely because the pandemic added additional barriers to physical activity [14,15,16,17,18,19,20,21,22,23,24,25,26,27,28]. PA further decreased after the onset of the COVID-19 pandemic at the beginning of 2020. The negative impact on PA opportunities for adolescents was one of the major consequences of the personal restrictions implemented in various countries to contain the virus’ spread. Important sources of PA, including schools, after-school structures, gyms, or sporting clubs, were not available, with a negative effect on PA levels. Due to local government-imposed restrictions, indoor activities decreased and outdoor activities (such as running or cycling) did not increase among adolescents [15,16]. Different tools had been used to confirm the data, including the Physical Activity Screening Measure, the Momo physical activity questionnaire, and a survey to verify the adherence to WHO guidelines of at least 60 min of PA a day [14,15,16]. The proportion of active adolescents during lockdown was in general low, in some cases ranging from 7.4% to 12.4% [15]. In some analyses, the percentage of those with an insufficient PA level dramatically increased, reaching a concerning threshold. For example, in Jordan, almost half of the subjects practiced at least 1 h of PA and in Australia, adolescent males were 88% less likely to meet guideline recommendations regarding MVPA and muscle-strengthening during lockdown [14,21]. Indoor activities, including fitness/aerobics or yoga/Pilates/stretching at home, were not commonly practiced by adolescents, mainly among the male gender [14]. In addition, some reports indicated that the changes were gender-related and that the decrease was significant in males [18]. Nevertheless, comparing PA levels, males were still more active during COVID-19 than females in many reports [22,23]. A possible explanation is that boys may engage in competitive sports more frequently than girls, but, during the pandemic, sports activity was often banned or restricted [18]. In line, a sufficient PA level has been mostly reported among adolescents with higher sports experience and achievement [20]. It is possible that adolescents who are more practiced in sports are capable of achieving appropriate PA even in situations where sports facilities and equipment are limited. Indeed, key factors for increasing adolescents’ PA include enjoyment, self-motivation, health perception, and gained expectations. Other authors did not find statistically significant differences in the variation of PA levels between boys and girls [19].

Another factor that may correlate with PA level is substance use, which includes tobacco, cannabis, and alcohol. In the literature, there was no validated correlation between PA and substance use, however, the evidence is conflicting. For example, Gilic B et al. reported a higher smoking prevalence in adolescents with an insufficient PA level during lockdown [20]. In contrast, a recent report by Chafee BW et al. did not confirm a significant variation from the pre-pandemic period and after the stay-at-home order, despite a relevant decrease in PA levels [17].

Weight gain among adolescents following COVID-19 onset was a direct result of PA decrease. Fromel K et al. reported that a significant increase in sitting time was noted compared to pre-pandemic times among adolescents. The percentage of those overweight/obese also increased, rising from 24.3% to 30.8% in boys and 12.9% to 14.1% in girls [19]. These results were confirmed by other authors, including Al Hourani H, who outlined an increment in obesity from 12.9% to 16.4% during the COVID-19 restriction. Eating habits were also not healthy, including eating more bread, french fries, popcorn, sugar, and ice cream, and consuming meals or snacks while watching television [21].

While physical inactivity increased, well-being decreased among adolescents during the pandemic period [15,19,27]. Using scales to capture stress symptomatology, including the Stress and Coping Questionnaire for Children and Adolescents or the WHO-5 Well-Being Index, well-being and mood health were investigated [15,19]. During the stay-at-home and distance education periods, adolescents reported a reduced percentage in a good level of well-being and increased symptoms of depression. A correlation between sadness, well-being, and PA had been evidenced in the literature, confirming the effect of COVID-19 restrictions on adolescents’ mental health [15,19]. In particular, adolescents with higher well-being scores reported less sedentary habits than their peers [19,22]. PA time was also related to sleep patterns during the lockdown, with an association found between insomnia, depressive syndrome, anxiety, and reduced PA among youth [24]. Increased evidence suggests that inadequate sleep quality or sleep duration is related to health problems in youth, including metabolic and cardiovascular diseases and psychological problems [29,30].

While PA levels were decreasing, an increase in the use of screen devices was noted [14,15,21,26]. Al Horani H et al. reported a significant increase in technology use after COVID-19 onset: 49.1% of adolescents spent more than 4 h a day on screens, compared to 22.4% in the pre-pandemic period [21]. Adolescents used technology not only for e-learning during school closure but also for recreational purposes [31]. Schmidt SCE et al. calculated an average of 67.8 more minutes per day spent on screen devices for recreational activity during the COVID-19 lockdown, including gaming and internet consultation [26]. Nevertheless, being connected by smartphone or tablet did not help them feel less lonely. On the contrary, depressive symptoms, anxiety disorders, and neuropsychological problems increased with the onset of the COVID-19 pandemic [10].

Evidence also suggests that parents may influence children’s PA and sedentary behavior. In Kesic et al.’s study, role modeling and reinforcements from parents proved to be one of the three main factors influencing PA levels, together with self-determination and availability of equipment [23,27]. The motivation of the adolescent could be a detrimental factor as well: the desire to remain fit, improve their physical appearance, or become an athlete was linked to PA during the pandemic [27]. Mothers’ education as well as parental care were linked to higher levels of PA among adolescents. Parents’ attitudes and values have a strong influence on adolescents’ behavior, and they can serve as role models in promoting PA [23]. PA should also be encouraged as part of the academic program as it may have a positive impact on attention, learning, and memory, without decreasing academic progress [32].

## 5. Conclusions

In conclusion, the main finding of the literature review was a decline in PA levels in adolescents during the pandemic, which emphasizes the need for substantial changes in the concept of PA. The fact that home PA did not increase and PA at school or after school was not replaced by any other form of exercise confirms the need to improve the awareness of the importance of PA and its benefits among youth. The risks of low PA levels include short-term (overweight, depression, and reduced well-being) and long-term impacts on health. In the event of a future pandemic or another unexpected life event, the following key factors for increasing PA levels in adolescents should be kept in mind:Increase education on the benefits of a regular PA and the risks of sedentary behavior;Promote support from family, friends, and teachers;Increase PA at school, as a part of the academic program;Increase availability of equipment and facilities to practice PA,Promote at-home PA options involving adolescents in informal exercise practice or online tutoring video/group fitness activities.

## Figures and Tables

**Figure 1 ijerph-20-03275-f001:**
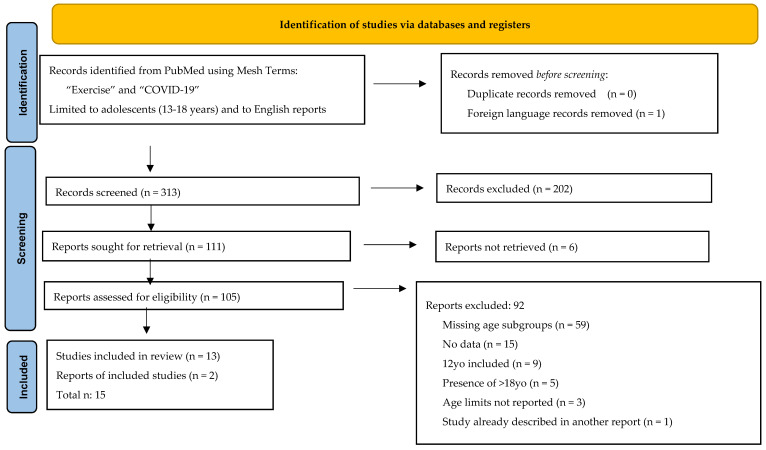
Flow chart of the selected process.

**Figure 2 ijerph-20-03275-f002:**
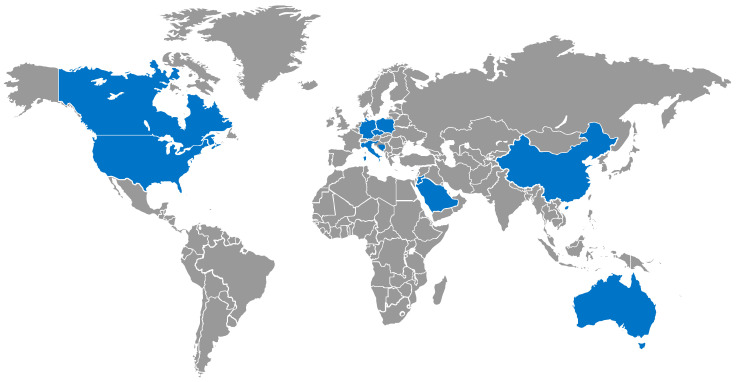
Countries involved in the studies are shown in blue on the world map; they are Australia, Bosnia and Herzegovina, Canada, China, the Czech Republic, Germany, Italy, Jordan, Poland, Saudi Arabia, and the U.S.A.

**Table 1 ijerph-20-03275-t001:** Main findings of the literature review. ^a^ MVPA: moderate-to-vigorous physical activity. ^b^ WHO guidelines: being active for at least 60 min every day in the week. ^c^ PAQ-A: Physical Activity Questionnaire for Adolescents. ^d^ mGodinLSI: modified Godin Leisure-Time Exercise Questionnaire Leisure-Time Score Index. ^e^ LPA: light physical activity.

Authors	Setting and Year of Publication	Type of Study	Participants (Age Range) Sex (F; M)	Type of Physical Activity (PA)	Findings
Arundell L et al. [14]	Australia, 2022	Survey	1296 participants (13–17 years) F = 903; M = 393	MVPA ^a^ and muscle strengthening	During the lockdown, adolescent males were 88% less likely to meet Australian guideline recommendations regarding moderate to vigorous physical activity (MVPA) and muscle strengthening.
Braksiek M et al. [15]	Germany, 2022	Survey	295 participants (14–17 years) F = 148; M = 147	Average active days meeting the WHO guidelines ^b^	The proportion of active adolescents during lockdown was low, ranging from 7.4% to 12.4%. Well-being and low levels of sadness positively correlated with PA; well-being was significantly lower than the average pre-pandemic status. No increased anger or higher anxiety was reported.
Bronikowska M et al. [16]	Poland, 2021	Survey	127 participants (14–16 years) F = 66; M = 61	MVPA ^a^	In total, 50% of active adolescents in the pre-pandemic time significantly decreased their level of MPVA below the WHO’s recommendations ^b^ (*p* = 0.01).
Chaffe BW et al. [17]	USA, 2021	Prospective cohort study	1006 participants (14–16 years) F = 623; M = 371; not specified = 12	PA for at least 20 min with sweat or heavy breathing (days/week)	PA decreased from 54% in pre-pandemic time to 38% after the stay-at-home order. No significant variations in the use of tobacco, cannabis, or alcohol were reported after restriction.
Elnaggar RK et al. [18]	Saudi Arabia, 2020	Survey	63 participants (14–18 years) F = 29; M = 34	Broad jump: explosive strength, sit-up: evaluated muscle endurance, sit-and-reach: assessed flexibility, and multilevel fitness: quantified aerobic endurance	PA levels decreased after COVID-19 onset. The changes were gender-related and the decrease was significant in males (*p* < 0.001).
Fromel K et al. [19]	Czech Republic and Poland, 2022	Survey	1349 participants (15–18 years) F = 723; M = 626	School PA, transportation PA, recreation PA, vigorous PA, moderate PA, walking, and overall Weekly PA	There was a significant decrease in PA during the pandemic. The most significant decrease, 30%, was reported in boys walking. Students with lower levels of well-being reported higher sedentary habits. Well-being decreased during the pandemic in both boys (from 65.6% to 50%) and girls (from 43.8% to 34.9%). Moreover, an increase in weight was noted: the percentage of overweight/obesity moved from 24.3% to 30.8% in boys (*p* < 0.01) and from 12.9% to 14.1% in girls.
Gilic B et al. [20]	Bosnia and Herzegovina, 2021	Prospective analysis	661 participants (15–18 years) F = 292; M = 369	Sufficient PA: PAQ-A ^c^ score > 2.73	PA significantly decreased during the COVID-19 pandemic, reaching an insufficient level in 48% of adolescents (versus 24% in the pre-pandemic period). The sufficient PA level significantly decreased from 67% to 37% in boys and from 28% to 9% in girls. Smoking cigarettes had a negative effect on PA levels.
Al Hourani et al. [21]	Jordan, 2021	Cross-sectional study	232 participants (13–17 years)	Playing time	During the lockdown, obesity increased from 12.9% to 16.4% and the time spent on a screen for more than 4 h a day increased to 49.1% from 22.4% at baseline. Almost half of the participants (42.7%) had no PA during the COVID-19 pandemic (vs. the pre-pandemic value of 19%).
Kang S et al. [22]	China, 2020	Survey	4898 participants (14–18 years) F = 2539; M = 2359	Total MVPA ^a^ divided into: vigorous, moderate, and walking. Sedentary time	After the onset of COVID-19, the MVPA of adolescents was 23 ± 52.5 min/day, with significantly higher sedentary levels in girls than in boys (*p* < 0.01). Girls also had higher scores of tension, anger, and confusion than boys (*p* < 0.01). Higher levels of PA were significantly associated with lower levels of negative mood scores (anger, fatigue, depression, and confusion) and higher positive mood scores (vigor and self-esteem) (*p* < 0.01).
Kesic GM et al. [23]	Bosnia and Herzegovina, 2021	Prospective analysis	859 participants (14–18 years) F = 371; M = 483	Sufficient PA: PAQ-A ^c^ score > 2.73	During the lockdown, sufficient PA levels significantly declined from 55% to 35% in the youngest, aged 14–16 years, and from 43% to 28% in those aged 16–18 years. Sufficient PA levels during lockdown were more likely in males (in the youngest group) and in the case of higher maternal education or parental care (in the older group).
Lu C et al. [24]	China, 2020	Cross-sectional study	965 participants (15–17 years) F = 409; M = 556	Physical activity time (PAT) (h/d): <1.5 h/d (low PAT) ≥1.5 h/d (high PAT) Sitting time (ST) (h/d): <4 h/d (low ST) ≥4 h/d (high ST)	In total, 49.9% of the sample size spent less than 1.5 h/day on PA. A low PA correlated with COVID-19 fear, insomnia, and depressive and anxiety symptoms (*p* < 0.01). Those with high PAT reported fewer opportunities to develop insomnia symptoms and depressive symptoms than those with low PAT (*p* < 0.05). In addition, compared to the group with unhealthy behaviors (low PAT + high ST), participants with healthy behaviors (high PAT + low ST) had a lower prevalence of insomnia (*p* < 0.001), depressive (*p* < 0.001), and anxiety symptoms (*p* < 0.05).
Minuto N et al. [25]	Italy, 2021	Retrospective study	57 participants (14–18 years)	No physical activity, PA < 3 h per week, and intense PA ≥ 3 h per week	A reduction in the weekly sports hours was observed (5.14 ± 4.20 at T= and 2.72 ± 3.40 at T1 *p* < 0.0001) during the COVID-19 lockdown and more adolescents became sedentary (14% at T0 versus 40.4% at T1).
Schmidt SCE et al. [26]	Germany, 2020	Survey	404 participants (14–17 years) F = 226; M = 178	Days active/week, PA guideline adherence ^b^, sports (minutes/day), and habitual activity (minutes/day): playing outside, walking and cycling, gardening, and housework	PA decreased (−21.1 min/day in males and −11.2 min/day in females) and recreational screen time increased (+79.2 min/day in males and +67.8).
Shepherd HA et al. [27]	Canada, 2021	Survey	20 participants (15–17 years) F = 10; M = 10	School extra-curricular sports	During COVID-19 restrictions, adolescents playing sports decreased their amount and intensity of PA, also reporting variations in social connections and mental health, with increased loneliness and anxiety. Prior to COVID-19 restrictions, PA helped them to manage their emotions, decreased their stress, and supported their physical and mental wellness.
Tulchin-Francis et al. [28]	USA, 2021	Survey	207 participants (14–18 years)	mGodinLSI ^d^: LPA ^e^, MVPA ^a^	PA scores declined significantly during the pandemic. Specifically, the MVPA score decreased to 25.6%. High stress levels had been reported but were not gender-related.

## Data Availability

Data available upon reasonable request to Dr. Bozzola.

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
