# Peer review of "Physical Activity in the COVID-19 Era and Its Impact on Adolescents’ Well-Being"

_ijerph, 2023, doi:10.3390/ijerph20043275_

Round 1

Reviewer 1 Report

Lines 39-46: You are making statements that are closely aligned to what you studied. Was this information from what the study found or from something else? It seems like this would be in the discussion not the introduction. Please clarify.

Lines 47-51: Need references for these statements.

Lines 188-190: Is this finding counter to what was believed or in line with such? The wording makes it unclear.

Concluding Thought

Overall, this was a solid paper that provides good information to the field. To see that PA was not replaced with at home activity when other affordances were taken away is to be noted, as we need to better prepare youth and adults for at home adherence to PA and exercise.  

Author Response

Dear Reviwer,

thank you for your notes. Please find attached an answer to your comments.

Reviewer 2 Report

The manuscript provides a study conducted with objective to review literature in order to analyze the changes in physical activity and exercise practice after the onset of COVID-19 pandemic and its impact on the well-being of adolescents. The study addresses a current and relevant issue in the field of health public. The manuscript is properly structured, the theoretical basis is satisfactory, and the references used are relevant to the subject of the study. However, with the intention of helping to improve the manuscript, I present some items that could possibly be adjusted.

(1) Introduction

Lines 25 to 27: Use the original reference equivalent to the definition of physical activity (Caspersen CJ, Powell KE, Christenson GM. Physical activity, exercise, and physical fitness: definitions and distinctions for health-related research. Public Health Rep. 1985; 100(2):126-31).

At some point in the introductory text differentiate the concepts of "physical inactivity" and "sedentary behavior".

(2) Materials and Methods – Exclusion Criteria

Lines 69 to 75: Delete the items “Reports in foreign languages different from English” and “Reports involving participants aged less than 13 or over 18 years old or for which results for subgroups were not available”. These are opposite items to the eligibility criteria.

(3) Results

Lines 158 to 160: Figure 1 (Countries involved in the studies) is not shown.

(4) Discussion

Lines 169 to 170: Change the phrase "In some analysis, the percentage of those who became sedentary dramatically increased, reaching concerning threshold" to "…., the percentage of those with an insufficient PA level dramatically increased, ……".

Author Response

Dear Reviewer,

thanks for your notes. Please, find attached a reply to the comments.

Reviewer 3 Report

Ccomments

It correspond to a review that sets the standard for planning interventions that contribute to improving the quality of life from childhood.

Although there is evidence of the relationship between well-being and the practice of physical activity, it is important for this study to identify the effect of physical activity on mental health. While the main question is to know the changes in the practice of physical activity and exercise after the COVID-19 pandemic and its impact on well-being in adolescents.

Although it is a descriptive study, a greater analysis of the variables could be included and an association established between them, mainly with regard to mental health as a key element for integral well-being, in this case of the population under study.

The results shown in Table 1. They describe the relevant information that answers the question and objective posed.  However, as a suggestion, the findings could be broadly described to emphasize the importance of physical activity and its relationship with comprehensive well-being in adolescents.
  The findings of Table 1 could be shown in a separate table to identify the differences by sex, by type of physical activity, by time and physiological condition for the case of depression and anxiety, to facilitate their reading.

Author Response

Dear Reviewer,

thank you for your notes and suggestion to improve the manuscript. Please, find attached a reply to your comments
